# DC-Gaussian: Improving 3D Gaussian Splatting for Reflective Dash Cam Videos

**Linhan Wang**[1][*] **Kai Cheng**[3][*] **Shuo Lei**[1] **Shengkun Wang**[1] **Wei Yin**[5]

**Chenyang Lei**[4] **Xiaoxiao Long**[2][†] **Chang-Tien Lu**[1]

[1]Virginia Tech [2]Hong Kong University [3]USTC [4]CAIR, HKISI-CAS [5]University of Adelaide

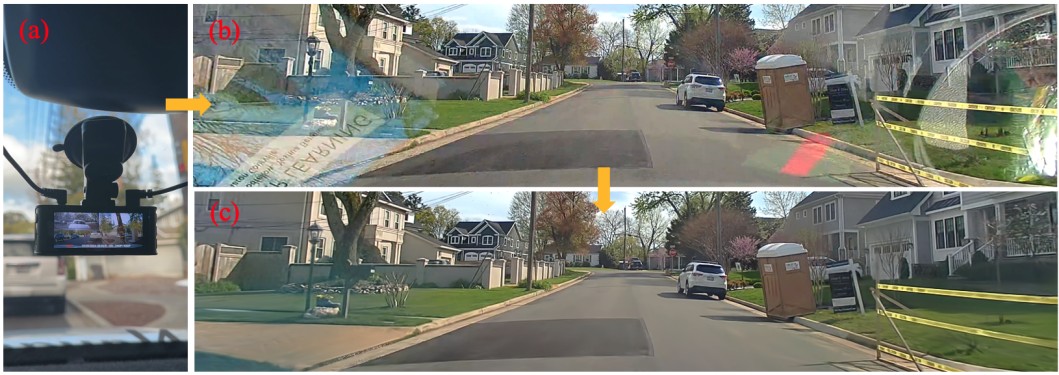

Figure 1: Given a sequence of video captured by a dash cam that may contain obstructions like reflections and occlusions, **DC-Gaussian** achieves high-fidelity novel view synthesis getting rid of the obstructions. (a) dash cam; (b) original video frame; (c) novel view rendering with obstruction removal.

## Abstract

We present DC-Gaussian, a new method for generating novel views from in-vehicle dash cam videos. While neural rendering techniques have made significant strides in driving scenarios, existing methods are primarily designed for videos collected by autonomous vehicles. However, these videos are limited in both quantity and diversity compared to dash cam videos, which are more widely used across various types of vehicles and capture a broader range of scenarios. Dash cam videos often suffer from severe obstructions such as reflections and occlusions on the windshields, which significantly impede the application of neural rendering techniques. To address this challenge, we develop DC-Gaussian based on the recent real-time neural rendering technique 3D Gaussian Splatting (3DGS). Our approach includes an adaptive image decomposition module to model reflections and occlusions in a unified manner. Additionally, we introduce illumination-aware obstruction modeling to manage reflections and occlusions under varying lighting conditions. Lastly, we employ a geometry-guided Gaussian enhancement strategy to improve rendering details by incorporating additional geometry priors. Experiments on self-captured and public dash cam videos show that our method

---

[*]Equal contribution
[†]Corresponding author

38th Conference on Neural Information Processing Systems (NeurIPS 2024).

not only achieves state-of-the-art performance in novel view synthesis, but also accurately reconstructing captured scenes getting rid of obstructions. See the project page for code, data: https://linhanwang.github.io/dcgaussian/.

# 1 Introduction

Neural Radiance Field (NeRF) [30] has revolutionized the image-based rendering area with its differentiable volume rendering technique. 3D Gaussian Splatting (3DGS) [19] pushes the frontier further with real-time rendering speed. These technologies have been applied to datasets captured by autonomous cars [42, 6, 25], opening up numerous new possibilities in autonomous driving, such as simulating driving scenarios [57, 51] for robust training of perception models and providing effective 3D scene representations to enhance comprehensive environmental understanding [14, 63, 66]. Although these datasets provide multi-modality sensor data, their diversity in real-world driving scenarios is still limited [7].

Fortunately, dash cam videos deeply reflect the diversity and complexity of real-world traffic scenarios [8]. They are used to provide large-scale, diverse driving video datasets in a crowd-sourced manner [59]. Dash cam videos also offer unique value by capturing multi-agent driving behaviors [7] and evaluating the robustness of algorithms under visual hazards [60]. More-over, the global dash cam market is rapidly expanding, driven by increasing awareness of vehicular safety [34]. Therefore, the exploration of utilizing dash camera data in neural rendering shows great potential, offering enormous amounts of data for autonomous driving applications.

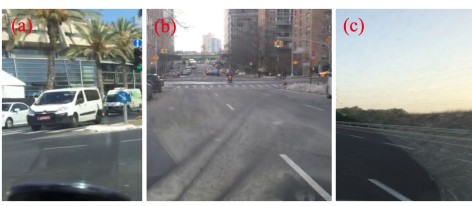

Figure 2: Common obstructions on windshields: (a) Mobile-phone holder; (b) Reflections; (c) Stains.

However, naively training 3DGS on dash cam videos often results in a significant deterioration of rendering quality and geometry. This degradation is primarily due to the common existence of obstructions (reflections and occlusions such as mobile phone holders and stains, as shown in Fig. 2) on windshields. In these scenarios, 3DGS models the obstructions as stationary geometries while they are dynamic in nature (moving with cars), thus unavoidably causing inaccurate geometry and blurry renderings in novel views.

Although some single-image-based obstruction removal methods exist, directly applying them to this task is nontrivial. These obstructions arise from various sources, while existing removal methods impose strict assumptions on obstructions [12, 64, 56, 55, 38, 20]. For instance, assumptions like out-of-focus [64] and ghost cue [38] allow previous methods to perform well in specific cases, but these assumptions don't always hold in dash cam videos. Moreover, the performance of learning-based methods degrades for out-of-distribution images [47, 27, 21]. Several NeRF methods [16, 33, 68, 69] attempt to reconstruct scenes with reflections by decomposing transmission (background scene) and reflection with independent NeRFs. This approach can benefit from the strong multi-view aggregation power of NeRF. However, previous methods are insufficient for dash cam videos because the obstructions on windshields do not align well with NeRF's design. The vanilla NeRF is intended for static scenes, whereas windshields and their reflected objects move with the cars.

In this paper, we introduce DC-Gaussian, a method for modeling high-fidelity obstruction-free 3D Gaussian Splatting from dash cam videos. We introduce three key innovations: 1) **Adaptive Image Decomposition**. To clearly decompose images with complex reflections and occlusions, we propose an adaptive image decomposition approach. We use an opacity map to learn the transmittance of the windshield, which adaptively estimates the contribution of the background scene to the image. 2) **Illumination-aware Obstruction Modeling(IOM)**. We observe that the obstructions are mainly caused by the objects being relatively static with the dash camera, but the obstructions present varying effects due to changing illumination. We therefore propose modeling global obstructions that are shared for all views. Moreover, a novel **L**atent **I**ntensity **M**odulation (**LIM**) module is introduced to learn the illumination changes from the scene and enable synthesis of reflection with varying intensity. 3) **Geometry-Guided Gaussian Enhancement(G3E)**. We further leverage multi-view

stereo to introduce geometry prior into 3DGS training, which enhances the details and sharpness of 3DGS rendering.

To evaluate the efficacy of our method, we conduct extensive experiments on public datasets [59] and a self-captured dataset. The experiments show that our method not only achieves state-of-the-art novel view synthesis, but also clearly removes obstructions from neural rendering.

## 2  Related Work

### 2.1  Novel View Synthesis for Driving Scenes

Novel view synthesis aims to render novel views given posed images of the same scene. NeRF [30], which combines multiple layer perceptions and differentiable volume rendering, initiated a revolution in this area by rendering photorealistic images. Subsequent works [3] [61] extended NeRF to unbounded large-scale scenes by warping the space into a bounded cube. BlockNeRF [44] first introduced NeRF to driving scenes by dividing the scenes into blocks and training separate models on them. The method of scene division was further improved in later works [45, 65]. To apply NeRF to multi-camera systems, UC-NeRF [10] addresses the under-calibration problem by refining the poses with spatio-temporal constraints. S-NeRF [52] uses sparse LiDAR points to enhance the training of NeRF and learn robust geometry. Decomposing dynamic objects and static backgrounds in driving scenes presents another challenge. Some works [46, 54] tackle this challenge with the help of LiDAR and 2D optical flows.

Recently 3DGS [19] has attracted great attention in the research community. It achieves optimal results in novel view synthesis and rendering speed by explicitly modeling a 3D scene with 3D gaussians. Some researchers have extended it to dynamic objects and scenes. Given a set of dynamic monocular images, a deformation network is introduced to model the motion of Gaussians [58]. DrivingGaussian [67] models and decomposes dynamic driving scenes based on composite gaussian splatting. GaussianPro [9] improves the geometry of 3DGS by controlling the densification process of 3DGS with classic PatchMatching algorithm [2]. Zhou et al. [66] propose to utilize 3D Gaussian Splatting for holistic urban scene understanding. However, dash cam videos, an important data source for understanding driving scenes, remain unexplored due to obstructions on the windshield. Despite previous works [67, 54, 58] making progress in separating dynamic objects from background scenes, the image decomposition problem we are addressing presents unique challenges because of the obstructions' transparent or semi-transparent nature.

### 2.2  Obstruction Removal and Layer Separation

**Single-image reflection removal.** To address the highly ill-posed problem of single-image reflection removal, various methods leverage different cues. Polarization cues are particularly valuable as they are inherently present in all natural light sources [36, 13, 21, 28]. Gradient priors [23, 24, 1] are utilized based on the observation that reflection and background layers often exhibit different levels of sharpness. Additionally, ghosting cues [38, 64, 18] and flash/non-flash pairs [20, 22] can be effective in certain scenarios. However, these assumptions do not always hold in real-world situations. With the advancement of deep learning technology, learning-based methods [17, 12, 50, 17] have been developed to model reflection properties more comprehensively. Despite their success, reflection removal from a single image remains challenging due to the inherently ill-posed nature of the problem, the absence of motion cues [27], and the difficulties in out-of-domain generalization [47].

**Multi-image layer separation.** Existing methods often exploit differences in motion patterns between transmission and obstruction layers and use learned image priors [15] to decompose images into multiple components. These layer separation methods estimate dense motion fields for each layer using optical flow [43], SIFT flow [40], and deep learning-based flow estimation methods [41, 27]. Recently, Nam et al. [32] propose a multi-frame fusion framework based on neural image representation, achieving strong performance on various layer-separation tasks. Similarly, NSF [11] fuses RAW burst image sequences by modeling optical flow with neural spline fields. However, methods designed for burst images struggle with large pixel motions in driving scenes.

**NeRF with reflections.** NeRFRen [16] is the pioneering work that adapts NeRF to model scenes with reflections by proposing to model transmitted and reflected components with separate NeRFs. NeuS-HSR [33] achieves high-fidelity 3D surface reconstruction by explicitly modeling the glasses

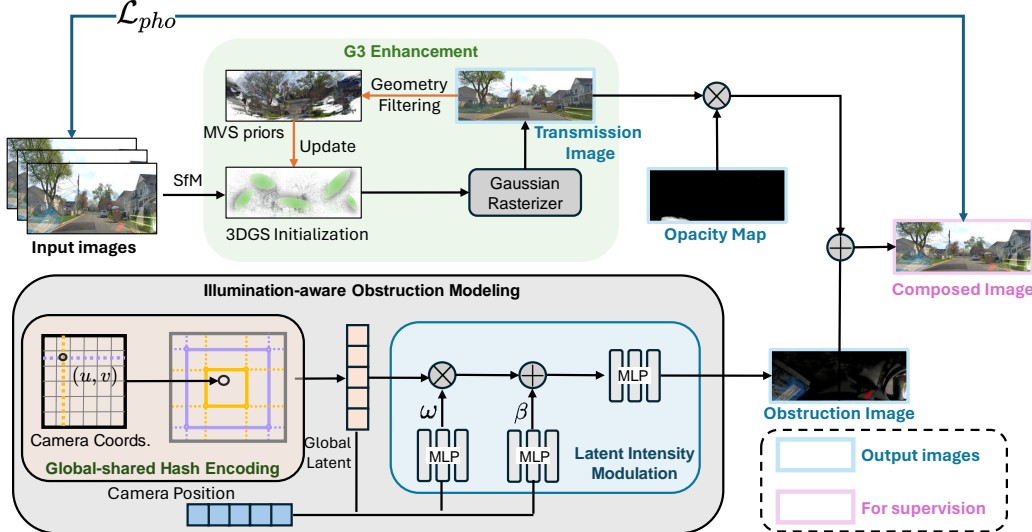

Figure 3: Overview of DC-Gaussian framework. To model obstructions with different opacities in a unified manner, we use an learnable opacity map to adaptively reweight the contribution of transmission. The global-shared multiresolution hash encoding is introduced to fully utilize the static motion prior of obstructions. We propose a Latent Intensity Modulation module to grasp the intensity changes of reflections conditioned on camera positions. Finally, in the G3 Enhancement module, we run geometry filtering on obstruction-suppressed images to enhance the geometry of 3D Gaussians.

with an auxiliary plane module. Zhu et al. [68] introduce recurring edge cues to achieve robust results under sparse views. However, previous methods are insufficient for dash cam videos because the obstructions on windshields do not align well with NeRF's design. The vanilla NeRF is intended for static scenes, whereas windshields and their reflected objects move with the cars. The varying illumination in the wild makes reflections modeling even more challenging. In contrast to existing methods, our proposed obstruction modeling approach leverages the static nature of obstructions in camera coordinates and captures the varying intensity of reflections.

## 3 Method

Our DC-Gaussian extends the standard 3DGS to corrupted dash cam videos. We begin by reviewing the standard 3DGS pipline 3.1. Then we introduce the Adaptive Image Decomposition 3.3 to decompose the reflections and occlusions from the corrupted dash cam images. In 3.3, we propose the Illuminate-aware Obstruction Modeling module. A novel Latent Intensity Modulation is introduced to enable high quality modeling of reflection even under vary illumination. Finally, the Geometry Guided Gaussian Enhancement strategy is explained in 3.4.

### 3.1 Preliminary of 3D Gaussian Splatting

3DGS [19] models the 3D scene as a set of 3D Gaussians. Each 3D Gaussian $G$ is defined as:

$$G(\boldsymbol{x}) = e^{-\frac{1}{2}(\boldsymbol{x}-\boldsymbol{\mu})^T \boldsymbol{\Sigma}^{-1}(\boldsymbol{x}-\boldsymbol{\mu})} \tag{1}$$

where $\boldsymbol{\mu}$ and $\boldsymbol{\Sigma}$ represent its mean vector and covariance matrix, respectively. For optimization purpose, the covariance matrix is further expressed as $\boldsymbol{\Sigma} = \boldsymbol{R}\boldsymbol{T}\boldsymbol{S}^T\boldsymbol{R}^T$, where $\boldsymbol{S}$ and $\boldsymbol{R}$ are the scaling matrix and rotation matrix, respectively. To render an image, the splatting technique [70] is applied. Specifically, the color of each pixel $\boldsymbol{p}$ is calculated by blending $N$ ordered Gaussians $\{G_i | i = 1, ..., N\}$ overlapping $\boldsymbol{p}$ as:

$$c(\boldsymbol{p}) = \sum_{i=1}^{N} \boldsymbol{c}_i \alpha_i \prod_{j=1}^{i-1} (1 - \alpha_j) \tag{2}$$

where $\alpha_i$ is obtained by evaluating a projected 2D Gaussian [70] from $G_i$ at $\boldsymbol{p}$ multiplied with a learned opacity of $G_i$, and $\boldsymbol{c}_i$ is the learnable color of $G_i$.

## 3.2 Adaptive Image Decomposition

To synthesize images with obstructions, previous methods [16, 33] render **transmission image** $I_t$ and **obstruction image** $I_o$ separately and utilize a naive linear combination of $I_t$ and $I_o$ to render the final output $I$, as illustrated in Eq. 3:

$$I = \phi_1 * I_t + \phi_2 * I_o, \tag{3}$$

where $\phi_1$ and $\phi_2$ are manually chosen. While this approach achieves descent performance on the pure reflection corrupted images, it cannot achieve good performance on dash cam videos with complex obstructions. Among the common obstructions, mobile-phone holders are opaque, stains is semi-transparent and reflections are transparent. Faced with this complex situation, inspired by [32], we propose to learn the opacity map $\phi$ from the input images. As a result, we reformulate the rendering process:

$$I(u, v, j) = (1 - \boldsymbol{\phi})I_t(u, v, j) + \boldsymbol{\phi} * I_o(u, v, j) \tag{4}$$

where $u, v \in [0, 1]$ are the continuous image coordinates and $j \in [0, 1, ..., N - 1]$ is the frame index. Instead of defining the opacity field in 3D world space, we define the opacity relative to the 2D image space of each view, resulting in a learnable 2D tensor in practice. This approach is more convenient for modeling obstructions because view-dependent effects are only related to the training images. In this image decomposition method, the transmission images represent the driving scenes viewed through the windshield. We can use standard 3DGS to model these scenes due to its multi-view consistent property. Thus, $I_t(u, v, j)$ can be easily calculated by Eq. 2, given the camera pose of frame $j$. $I_o(u, v, j)$ represents the appearance of the complex obstructions on the windshield. In the implementation, we incorporate $\phi$ into $I_o(u, v, j)$ in the second term of Eq. 4 for robust training. We explain the modeling of $I_o(u, v, j)$ in the next section.

## 3.3 Illumination-aware Obstruction Modeling

Decomposing the images into transmission and obstruction is challenging due to the strong ambiguity between the two components. It is an ill-posed problem without prior information. We have two observations about the obstructions on the windshield that can strongly mitigate this ambiguity.

**Observation 1** As shown in Fig. 4 (a), the reflections on the windshields are from objects (air conditioner vent) inside the car, which means they are relatively stationary with the car [39]. Additionally, occlusions are attached to the windshields, which are also relatively stationary with the car. Consequently, **we can assume that the appearance of the obstructions is like an image shared globally by all the frames in a dash cam video**.

**Observation 2** As cars move along the road, the trees and buildings on the sides occasionally block the incident light, affecting the intensity of the reflections. For example, under strong light, reflections are also strong, as shown in Fig. 4 (a), while the reflections "disappear" under weak light, as seen in Fig. 4 (d). Thus, **the strength of the reflections is conditioned on the car's position**.

These two observations require us to design a model for obstructions that takes advantage of the global-sharing property of obstructions and also grasps varying intensity reflections conditioned on the car's position.

**Global-shared Multi-resolution Hash Encoding.** To align our design with Observation 1, we use a global-shared latent representation for obstructions' appearance. Specifically, we use continuous image coordinates $u, v \in [0, 1]$ as the input. Then we use a multi-resolution hash encoder $\gamma$ [31] to map these coordinates into high-dimensional learnable latent features. For example, we use an $L$

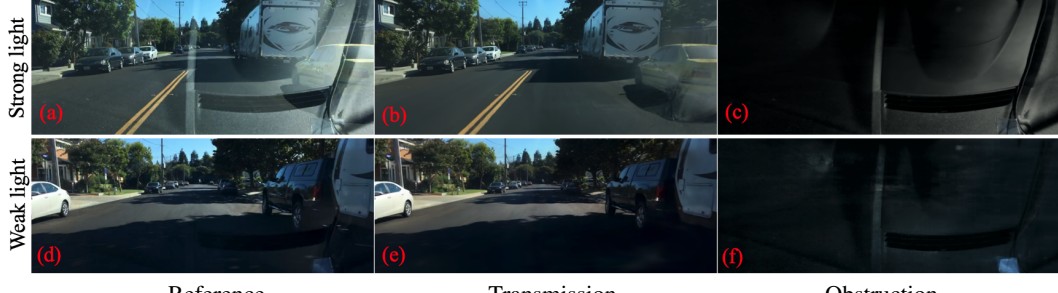

| Reference | Transmission | Obstruction |

Figure 4: When the intensity of incident light changes, the strength of reflections also changes accordingly (a, d). Our method achieves high-fidelity reflections synthesis (c, f) and reasonable decomposition results (b, e) under varying light. The reflections in (f) are too weak to be seen by the eye, so we brighten it to reveal the details.

level hash encoder where each level stores $F$ dimensional features. Thus, each $u, v$ pair maps to an $L * F$ dimensional latent features $\gamma(u, v)$. While our method is not restricted to this specific type of spatial encoding, we choose the multi-resolution hash encoder for two reasons. First, its hierarchical multi-resolution representation can adaptively learn the obstruction appearance in a coarse-to-fine fashion. Second, its efficient implementation matches the speed of 3DGS, without slowing down the training significantly.

**Latent Intensity Modulation.** In order to accurately capture the intensity variations in environmental lighting conditioned on car positions, we propose a novel Latent Intensity Modulation (LIM) module. Specifically, to enable selective activation of reflections conditioned on camera positions, we design a *Scaling Gate* $\omega$ and an *Offset Gate* $\beta$, which are generated by two MLPs with the concatenation of camera positions $\boldsymbol{\pi}_j$ and $\gamma(u, v)$ as input.

$$\omega(u, v, \boldsymbol{\pi}_j) = \mathrm{MLP}([\boldsymbol{\pi}_j, \gamma(u, v)]), \beta(u, v, \boldsymbol{\pi}_j) = \mathrm{MLP}([\boldsymbol{\pi}_j, \gamma(u, v)]), \tag{5}$$

Then, the latent features are modulated element-wise through these two gates. Finally, we use a MLP to decode the modulated latent features into RGB information of obstructions.

$$I_o(u, v, \boldsymbol{\pi}_j) = \mathrm{MLP}(\omega(u, v, \boldsymbol{\pi}_j) \odot \gamma(u, v) + \beta(u, v, \boldsymbol{\pi}_j)) \tag{6}$$

With the LIM module, our method achieves effective image decomposition and synthesizes high-fidelity reflections under varying light conditions, as shown in Fig. 4.

### 3.4 Geometry-Guided Gaussian Enhancement

In IOM, we utilize the motion pattern prior of obstructions to reduce ambiguity in decomposing images. Despite this, some ambiguity still remains. To further enhance performance, we incorporate geometry priors. Typically, strong obstructions affect only portions of the images. As cars move, the same objects in the 3D world can appear in several image fragments which are not or less interfered with by obstructions. In these image fragments, the texture in the transmission is less blurred by obstructions, and multi-view consistency is maintained. Based on this intuition, we leverage a multi-view stereo (MVS) algorithm to identify these "image fragments" and generate geometry priors for 3D Gaussians.

Specifically, we employ a deep MVS method [48] to generate depth maps for all views. A geometric consistency filtering process [37] is leveraged to generate masks $M_j$ masking the multi-view consistent areas, which are essentially the "image fragments" we want to identify. The masked depth maps are then mapped to 3D space as dense point clouds, which are used to initialize 3D Gaussians at physically accurate positions. To find more multi-view consistent "image fragments," we propose suppressing the obstructions in training images by employing our proposed method. Specifically, we remove obstructions from input images to obtain transmission $\hat{I}_t^j$. For areas blocked by occlusions(where $\phi$ has large value), we inpaint the content with $I_t^j$.

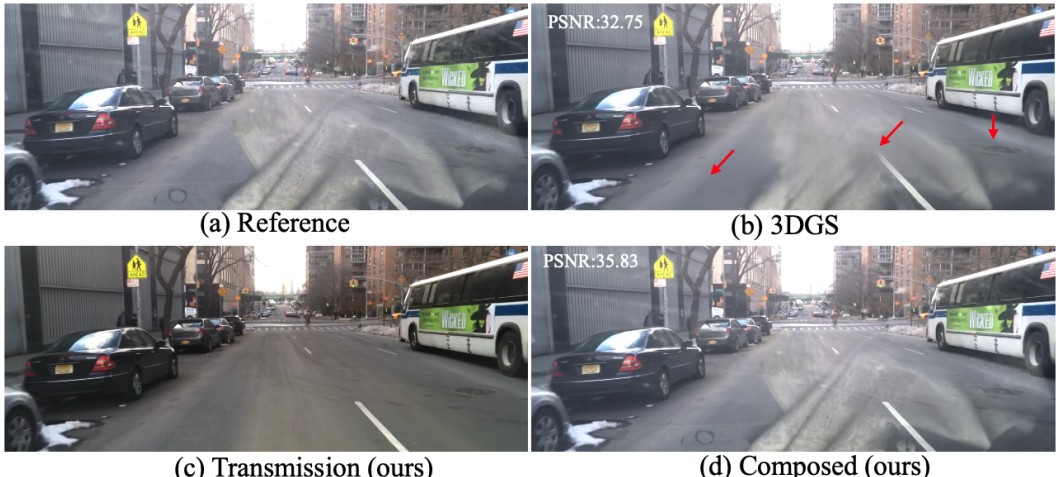

|  |  |
|---|---|
| (a) Reference | (b) 3DGS |
| (c) Transmission (ours) | (d) Composed (ours) |

Figure 5: Comparisons with 3DGS on novel view synthesis. Because the obstructions violate multi-view consistency, the performance of 3DGS degrades significantly, resulting in artifacts and blurry renderings (highlighted by red arrows). In contrast, our method not only faithfully synthesizes novel view renderings but also renders transmission with fine details, exhibiting an improvement of 3.05dB in terms of PSNR.

$$\hat{I}_t^j = \begin{cases} (I^j - I_o^j)/(1 - \phi), & \text{if } \phi < \tau \\ I_t^j & \text{otherwise} \end{cases} \tag{7}$$

Here $I^j$ is the $j_{th}$ input image. $I_o^j$ and $I_t^j$ are synthesized by trained IOM and 3DGS, respectively. Eq. 7 is derived from Eq. 4. We use 0.5 for the threshold $\tau$ in all the experiments.

### 3.5 Implementation Details

We develop our method based on 3DGS. We borrow multi-resolution hash encoding and fast MLP implementation from tiny-cuda-nn [31] to build IOM. We choose PatchMatchNet [48] as the MVS method in G3E. We follow previous driving scenes reconstruction works [46, 10] to separately model the sky areas. The final loss function is:

$$\mathcal{L} = \mathcal{L}_{pho} + \lambda_1 \mathcal{L}_{sky} + \lambda_2 \mathcal{L}_{opacity} \tag{8}$$

$\mathcal{L}_{pho}$ is the same as in standard 3DGS [19]. $\mathcal{L}_{sky}$ is borrowed from UCNeRF [10]. We use a $L_1$ loss $\mathcal{L}_{opacity} = \sum_{(u,v)} \|\phi(u,v)\|_1$ to regularize the opacity field, where $(u,v)$ are the image coordinates. This opacity loss encourages the opacity map to have the minimum areas that could satisfy the optimization. This design is based on the prior knowledge that opaque objects typically occupy only small portions of the windshield. We use 0.001 for both $\lambda_1$ and $\lambda_2$. We run all the experiments with an A100 GPU. Each scene contains approximately 300 images in our evaluation datasets. Combined training of the 3DGS and IOM for 30k iterations with Adam optimizer takes about 30 minutes. It takes 40 minutes in total to evaluate MVS and run geometry filtering.

## 4 Experiments

### 4.1 Datasets

**BDD100K.** To evaluate the performance of our method and baselines, we adopt BDD100K [59] for evaluation. This dataset contains 8 scenes that are from dash cam videos captured in daily life. They contain common obstructions, such as reflections, mobile phone holders, stickers and stains. Evaluation on this dataset reflects performance on real life dash cam videos.

Table 1: Evaluation of novel view synthesis on BDD100K and DCVR. We indicate the best and second best with bold and underline respectively. Our method consistently outperforms state-of-the-art methods in both datasets and all the evaluation metrics.

| Method | BDD100K | | | DCVR | | | FPS ↑ |
|---|---|---|---|---|---|---|---|
| | PSNR ↑ | SSIM ↑ | LPIPS ↓ | PSNR ↑ | SSIM ↑ | LPIPS ↓ | |
| NeRF-W [29] | 22.58 | 0.708 | 0.395 | - | - | - | 0.18 |
| ZipNeRF [4] | 27.89 | 0.875 | 0.176 | 24.41 | 0.786 | 0.228 | 0.27 |
| GaussianPro [9] | 27.75 | 0.894 | 0.192 | 23.71 | 0.770 | 0.270 | **210** |
| 3DGS [19] | 28.02 | 0.897 | 0.188 | 23.73 | 0.783 | 0.248 | 155 |
| DCGaussian (Ours) | **29.44** | **0.914** | **0.143** | **24.74** | **0.822** | **0.202** | 120 |

**DCVR.** To further evaluate the performance of our method on strong reflection conditions, we established **DCVR** (Dash Cam Videos with Reflection) dataset. This dataset contains 10 dash cam videos we collect. The original videos are undistorted [5] to ease the structure-from-motion algorithm. We utilize the popular tools COLMAP [37] and HLoc [35, 26] to estimate the camera parameters.

In both datasets, each sequence consists of approximately 300 frames, extracted from 10-second videos at a frame rate of 30 Hz. For scenes containing car hoods, the lower parts of the images are removed during preprocessing. Seven out of eighteen scenes in two datasets involve car turns, introducing diverse illumination changes. Additional details and visual results are provided in the appendix.

## 4.2 Baselines

We choose 3DGS as our baseline [19]. We also compare our method with other state-of-the-art methods Zip-NeRF [4], NeRF-W [29] and GaussianPro [9]. We use the unofficial implementations of Zip-NeRF [3] and NeRF-W [4]. To evaluate the performance of novel view synthesis, following common settings [3], we select one of every eight images as testing images and the remaining ones for training. Since our method is also designed for image decomposition, we also compare our method with state-of-the-art obstruction removal methods, including DSRNet [17], NIR [32] and Liu et al. [27].

## 4.3 Quantitative Results

For the quantitative evaluation, we conduct comparison with baselines on both BDD100K and DCVR. We apply the three widely-used metrics for evaluation, i.e., PSNR, SSIM [49], and LPIPS [62]. The results are shown in Tab. 1. The consistent superior performance of our method shows the efficacy of the proposed modules. Though GaussianPro and Zip-NeRF achieve great performance in obstruction-free scenes with their progressive propagation strategy and anti-aliasing mechanism, without separate obstruction modeling, they cannot handle the obstructions corrupted dash cam videos. NeRF-W is designed to handle illumination and content difference between images taken at different times but still cannot handle obstructions on the windshield. We show more visual results in the appendix. DC-Gaussian achieves 120 fps at a resolution of 1920x1080 on an RTX 3090 GPU. Although ours is slightly slower than 3DGS, the speed still enables real-time rendering, which is crucial for applications such as autonomous driving simulators.

## 4.4 Qualitative Results

**Novel view synthesis.** We show the novel view synthesis results in Fig. 5. Without a proper representation for obstructions, 3DGS can hardly synthesize high-quality obstructions. Moreover, its wrong geometry also results in blurry renderings and artifacts on the road surface. In contrast, our method effectively tackles the ambiguity between obstructions and transmissions and synthesizes both components with high-fidelity. We provide depth map in the appendix.

**Obstruction removal.** We show the obstruction removal results in Fig. 6. The single image reflection removal method (e) only marginally suppresses reflections. Multi-image layer separation methods

---

[3]https://github.com/SuLvXiangXin/zipnerf-pytorch
[4]https://github.com/kwea123/nerf_pl

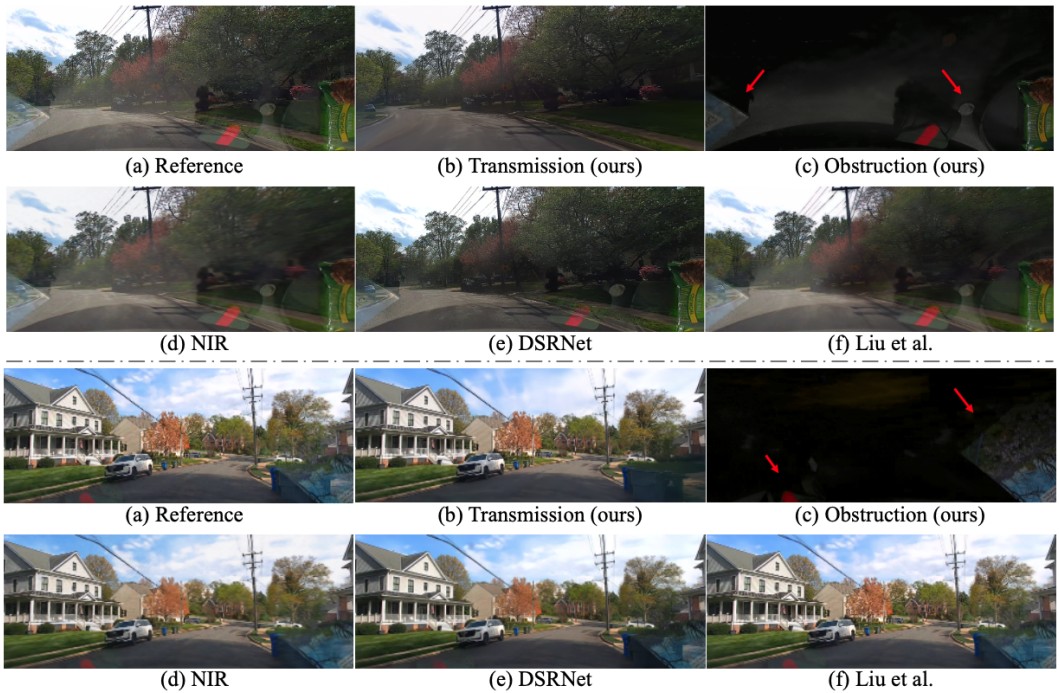

(a) Reference     (b) Transmission (ours)     (c) Obstruction (ours)

(d) NIR     (e) DSRNet     (f) Liu et al.

(a) Reference     (b) Transmission (ours)     (c) Obstruction (ours)

(d) NIR     (e) DSRNet     (f) Liu et al.

Figure 6: Comparisons with the image-based reflection removal methods on removing reflections. (a) Reference is a frame in a dash cam video. (b) and (c) are transmission and obstruction decomposed by our method. (d), (e), and (f) are results from previous obstruction removal methods, NIR [32], DSRNet [17] and Liu et al. [27], which are not effective in this scenario. In comparison, our method decomposes the image and synthesizes (b) transmission and (c) obstruction with high fidelity.

Table 2: Ablations studies on DCVR. Metrics are calculated on obstruction influenced areas.

| NOM | AD | LIM | G3E | PSNR ↑ | SSIM ↑ | LPIPS ↓ |
|---|---|---|---|---|---|---|
| ✗ | ✗ | ✗ | ✗ | 23.99 | 0.738 | 0.287 |
| ✓ | ✗ | ✗ | ✗ | 25.21 | 0.776 | 0.252 |
| ✓ | ✓ | ✗ | ✗ | 25.65 | 0.791 | 0.236 |
| ✓ | ✓ | ✓ | ✗ | 25.90 | 0.798 | 0.229 |
| ✓ | ✓ | ✓ | ✓ | **26.30** | **0.814** | **0.210** |

(d)(f) struggle with accurate optical flow estimation, resulting in blurry outputs. In comparison, our method models reflections (c) with high quality and retains fine details in transmission (b). These visual results demonstrate our method's potential in reconstructing obstruction-free driving scenes from dash cam videos.

## 4.5 Ablation Study

We conduct extensive ablation studies on DCVR to explore the impact of each module in DC-Gaussian. Quantitative results are shown in Table 2. To assess the efficacy of global-shared hash encoding, we evaluate 3DGS with a Naive Obstruction Module (NOM), where latent features are directly decoded by an MLP to generate RGB. NOM shows significant improvement over the baseline, demonstrating that global-shared hash encoding effectively leverages the static prior of obstructions in dash cam videos. The adaptive image decomposition (AD) strategy further enhances results by effectively modeling occlusions, as shown in Fig. 7. Additionally, LIM improves performance by capturing intensity changes in reflections. Finally, incorporating geometry priors into 3D Gaussians with G3E effectively suppresses artifacts on the road and reveals sharper details, as illustrated in Fig. 8.

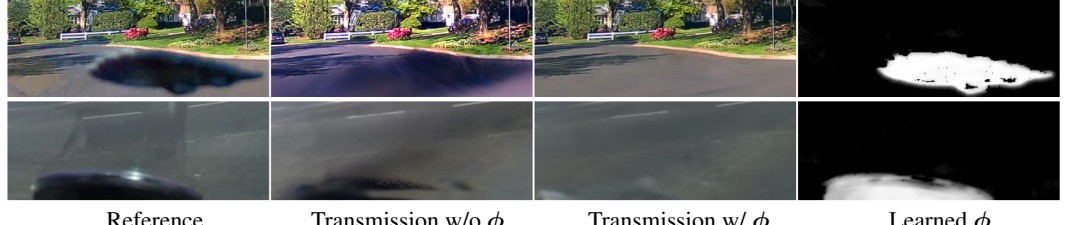

| Reference | Transmission w/o $\phi$ | Transmission w/ $\phi$ | Learned $\phi$ |

Figure 7: Ablation study about the Learnable opacity map $\phi$. Incorporating the opacity map allows our method to accurately identify the positions of opaque objects, enhancing physical simulation and improving view synthesis and obstruction removal. Without the opacity map, severe artifacts appear.

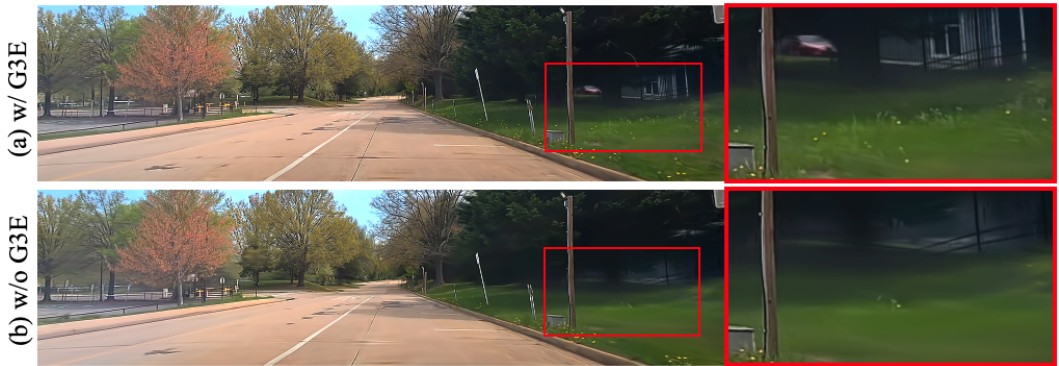

Figure 8: Ablation study on G3E module. G3E helps suppress artifacts and reveal sharper details.

## 5  Conclusions

In conclusion, we propose DC-Gaussian, which effectively addresses the challenges of extending 3D Gaussian Splatting to dash cam videos for the first time. The proposed Adaptive Image Decomposition module enables unified modeling of reflections and occlusions. To handle the reflections and occlusions under challenging lighting conditions, we introduce Illumination-aware Obstruction modeling. Additionally, we employ a Geometry-Guided Gaussian Enhancement strategy to further improve rendering quality. Experiments on BDD100K and DCVR demonstrate significant improvements in rendering quality and image decomposition, setting a new benchmark for neural rendering with dash cam videos.

**Limitations and future work** Currently, DC-Gaussian has only been evaluated on single-sequence videos. However, considering the vast amount of dash cam footage available, extending DC-Gaussian to a multi-sequence video setting and leveraging dense view images to achieve more pleasing results would be a promising direction for future research. In addition, Our method is not specifically designed to improve performance on dynamic scenes. We provide additional experimental results in the appendix. The results demonstrate that dynamic objects do not significantly impact the performance of obstruction removal. When dynamic objects move at a slow speed, our method also presents reasonable results. We plan to incorporate techniques [53] for dynamic objects modeling into our method in future research to enable robust dynamic modeling.

## Acknowledgments and Disclosure of Funding

The authors acknowledge Advanced Research Computing at Virginia Tech for providing computational resources and technical support that have contributed to the results reported within this paper. URL: https://arc.vt.edu/

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

# A  Appendix / supplemental material

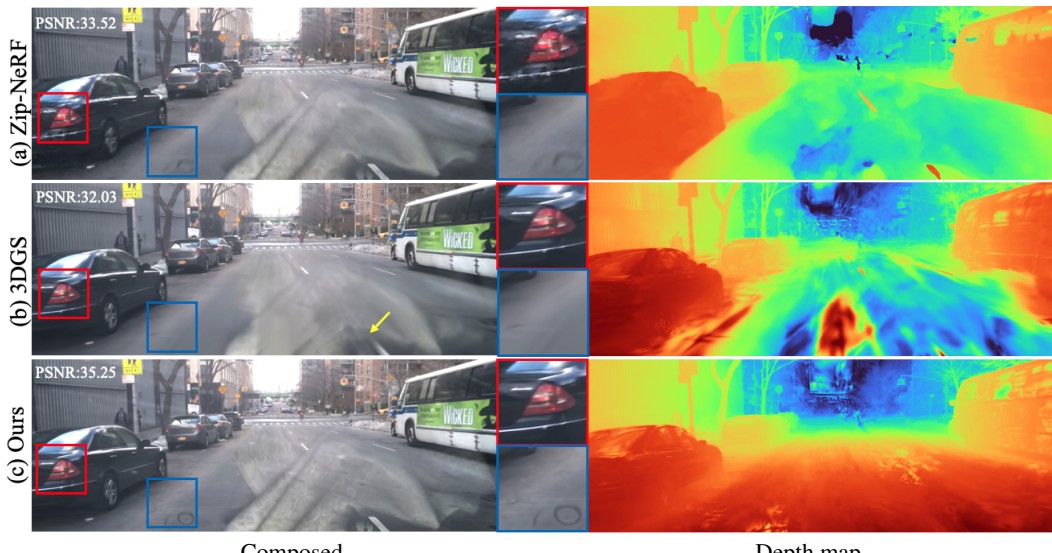

Figure 9: Comparisons with 3DGS and Zip-NeRF [4] on novel view synthesis show that obstructions violate multi-view consistency, leading to erroneous geometry in 3DGS and Zip-NeRF, as evident in the depth maps. This results in blurry renderings and artifacts. In contrast, our method effectively addresses the ambiguity introduced by obstructions and learns physically reasonable geometry, achieving renderings with fine details.

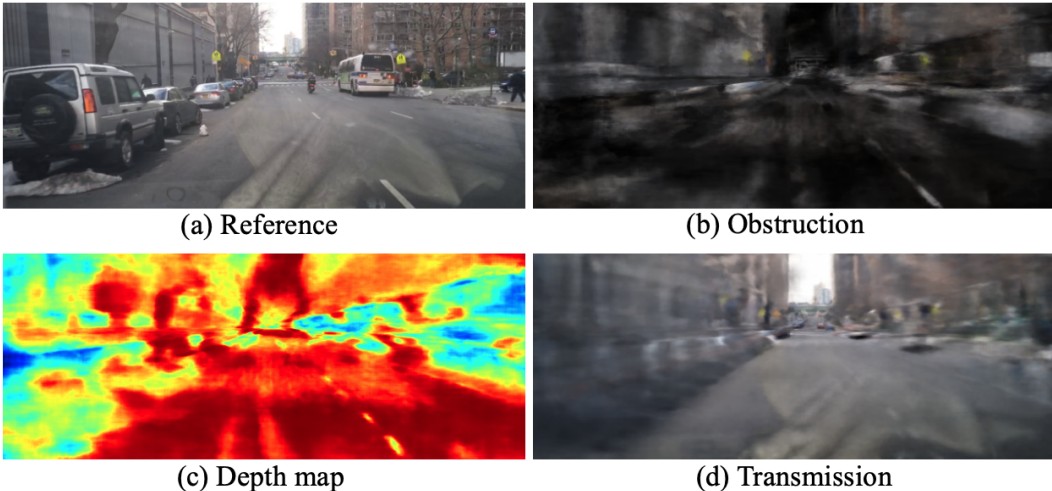

Figure 10: We evaluate NeRFRen [16] on our curated dataset. The suboptimal results of NeRFRen are caused by two factors. First, its obstruction modeling cannot address the ambiguity between obstructions and transmission, leading to a failure in image decomposition. Second, its backbone, NeRF, cannot handle large-scale driving scenes, resulting in blurry outputs.

Table 3: Ablation on threshold $\tau$ used in Eq. 7. Our results are not sensitive to the choice of $\tau$.

| $\tau$ | PSNR ↑ | SSIM ↑ | LPIPS ↓ |
|---|---|---|---|
| 0.3 | 26.27 | 0.814 | 0.210 |
| 0.5 | 26.30 | 0.814 | 0.210 |
| 0.7 | 26.28 | 0.814 | 0.210 |

Table 4: We first use DSRNet [17] to remove reflections from the input images, and then we train and evaluate 3DGS [19] on these images. The results show that due to the insufficiency of the reflection removal, novel view synthesis performance cannot be improved in this way.

| Method | BDD100K | | | DCVR | | |
|---|---|---|---|---|---|---|
| | PSNR ↑ | SSIM ↑ | LPIPS ↓ | PSNR ↑ | SSIM ↑ | LPIPS ↓ |
| 3DGS | 28.02 | 0.897 | 0.188 | 23.73 | 0.783 | 0.248 |
| 3DGS + DSRNet | 27.99 | 0.898 | 0.188 | 23.72 | 0.783 | 0.248 |

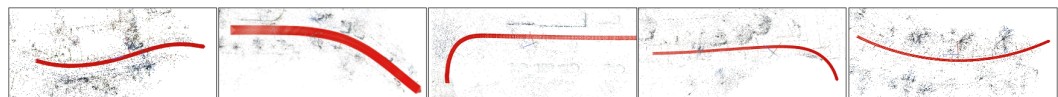

Figure 11: Trajectories of turning cars, which result in diverse illumination changes.

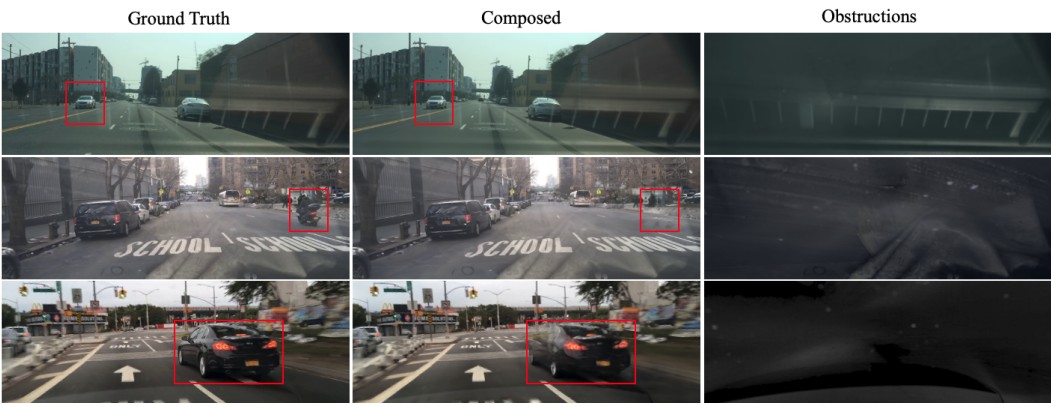

Figure 12: Visual results on dynamic scenes. All three scenes demonstrate that dynamic objects do not significantly impact the decomposition of obstructions. Our method achieves good performance in the scene shown in the first row, where the dynamic objects are moving slowly. However, in the second and third rows, where the dynamic objects are moving at higher speeds, our method shows suboptimal performance.

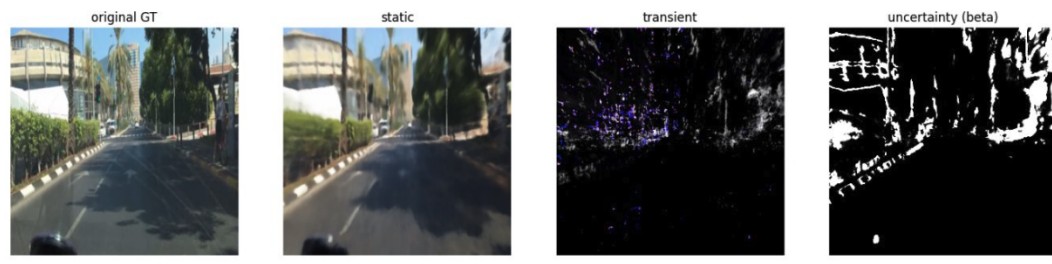

Figure 13: Nerf-in-the-wild fails to separate obstructions from the images. None of the obstructions are accurately represented in the transient image.

