# OpenReview forum: "DC-Gaussian: Improving 3D Gaussian Splatting for Reflective Dash Cam Videos"
_NeurIPS.cc/2024/Conference — NeurIPS 2024 poster_

### Official Review · Reviewer_QCPG · 2024-06-24

**Soundness:** 3
**Presentation:** 3
**Contribution:** 3
**Rating:** 6
**Confidence:** 5

**Summary:**

This paper tries to reconstruct 3DGS (3D Gaussian Splatting) with Dash Cam Videos, which introduces obstructions on windshields. An adaptive image decomposition is applied to learn transmission images and obstruction images separately. The former is modeled with 3DGS in G3 Enhancement and are rendered into 2D space, while the latter is learned on 2D camera space conditioned by car positions in Illumination-aware Obstruction Modeling. Further, an opacity map is optimized to weigh these two images. This method surpasses Zip-NeRF, 3DGS and other runner-ups in public and self-collected datasets.

**Strengths:**

Originality: This paper addresses the issue faced in daily life where perfect capturing is hard to get. The setting of Dash Cam Videos is novel, and I'm happy to see the introduction of 3DGS to model such straight-ahead-captured data.

Quality: The results seem good, with high scores on rendering and outstanding decomposition on transmission and obstruction.

Clarity: I believe the paper and diagrams are easy to read.

Significance: This may do help to autonomous driving considering its high-speed rendering.

**Weaknesses:**

1. The opacity map is modeled to be independent of car position, but in fact, according to Fresnel's law, opacity is also related to the light distribution, and therefore also related to the position.
2. The two observations in 3.3 are too strong. This somehow weakens the generalization of the algorithm.

**Questions:**

1. How to calculate opacity loss in Eq.8?
2. Are there any visualizations related to reconstructed 3DGS?
3. How fast is the rendering?

**Limitations:**

As stated in this paper, DC-Gaussian has only been evaluated on single-sequence videos. I would also prefer the opacity map to be detailed modeled. However, considering the good performance, this trade-off on opacity is acceptable.

---

> ### Author Rebuttal · Authors · 2024-08-06
>
> **1) Why is the opacity map modeled to be independent of car position?**
>
> Thank you for pointing out that, according to Fresnel's law, the opacity of the window should also depend on the car's position.
> However, in experiments, we found that most of occlusions are caused by the objects between dash camera and car window, such as mobile phone holders and stains, rather than the objects outside the car. Since these objects remains relatively still to cars, it is reasonable to design the opacity map to be independent of the car's position. Given the extremely ill-posed nature of this task, we chose to omit the dependence between window's opacity and car's position.
>
> **2) The two observations in 3.3 about the static motion prior and vary illuminations of obstructions are too strong.**
>
> We acknowledge that the assumptions guiding our design are quite strong. However, this task is highly ill-posed, as evidenced by the suboptimal results of previous methods, such as some reflection removal models trained on paired datasets. These strong assumptions have been proved to be crucial for achieving satisfactory performance.
> We will explore to relax these assumptions in the future research, enabling adaptation to a wider range of scenarios.
>
> **3) Details of $\boldsymbol{\mathcal{L}_{opacity}}$.** This question is answered in the global response.
>
> **4) Rendering speed.** This question is answered in the global rebuttal response.
>
> **5) Visualization of 3DGS.** We provide a visualization of the generated 3DGS in Figure 1 of the attached PDF. Due to the one-page limit of the attached document, please refer to our supplementary materials for additional visual results.

---

> > ### Comment · Reviewer_QCPG · 2024-08-11
> >
> > After thoroughly reading the PDF and the rebuttal, I would like to thank the author for adequately addressing my concerns. I will maintain my opinion that this is a very interesting setting. Since exposed cameras are highly susceptible to physical damage, I believe Dash Cam Videos present a potential solution. At least they could provide valuable prior knowledge fusion in automatic driving.
> > As for the inaccuracies in opacity map modeling, there is no significant degradation in the results, so I'm willing to accept this trade-off. However, I hope the author seriously considers this issue in future work.
> > I'll keep my score.

---

> > > ### Author Response · Authors · 2024-08-11
> > > **Thanks for the Response**
> > >
> > > Thank you so much for your valuable feedback. We greatly appreciate your acknowledgment of this setting and your insightful suggestions for improving the modeling of the opacity map. We will conduct more experiments and analysis on the opacity map in our future work.

---

### Official Review · Reviewer_zoS8 · 2024-07-06

**Soundness:** 3
**Presentation:** 4
**Contribution:** 3
**Rating:** 5
**Confidence:** 5

**Summary:**

This paper presents a new Gaussian Splatting novel view synthesis method specifically for in-vehicle dash cam videos containing various obstructions. To the challenges caused by obstructions, the proposed method separately represents the transmission and obstruction part of the camera capture. The transmission part is a 3DGS representation, and the obstruction part is modeled as a 2D representation with an opacity map and position-dependent illumination-aware neural model. The synthesized transmission images are also used to enhance the geometry prior for 3DGS training. The evaluation results show apparent improvements over previous methods.

**Strengths:**

1. The paper has an interesting motivation for a neglected but meaningful task on driving scene reconstruction with the dash cam. The paper is well-written and easy to follow. The paper also gives adequate references to related work.
2. The proposed solution looks reasonable and very effective on this specific task. I love to see how the authors leverage the key observations to design their solutions.
3. The latent intensity modulation (LIM) is an interesting and smart design for handling varying illumination conditions on the road.

**Weaknesses:**

1. The proposed method is more like an improvement for a specific Gaussian Splatting application. The similar components in the proposed method can be found, in part, from other related work. Therefore, the novelty of this paper is moderate.

2. In Sec. 3.2, the authors reformulate the composition equation to Eq. 4. This equation is a bit incorrect from the perspective of light transport. I feel the $(1 - \phi) I_t + (\phi) I_o $ would make more sense.

3. In the experiments, it is still unclear about the evaluation datasets. The following key informations are missing in the paper:

   * The number of scenes used and the number of frames per scene.
   * Image quality, and resolutions (since dash cam might have worse image quality).
   * Do the scenes contain dynamic objects?
   * Is there any scene containing opaque occlusions such as a car hood?
   * Do the drive sequences contain any turning to show more diverse illumination changes?

   I would suggest the authors show more visual results in the revised version.

I am looking forward to the authors' responses.

**Questions:**

* I wonder how well the proposed method works for the reflective car hoods, which are also commonly seen from the dash cam.

**Limitations:**

Limitations are discussed at the end of the paper, but I don’t think it is adequate, see my comments in the above sections.

---

> ### Author Rebuttal · Authors · 2024-08-06
>
> **1) Moderate novelty because the similar components in the proposed method can be found, in part, from other related work.**
>
> We appreciate if the reviewer could be more specific about the components. Then we can provide detailed explanations during the discussion period.
> While both 3DGS and hierarchical hash encoding are tools that have been developed and widely used in related works, the novelty of this paper lies in our effective design of a framework that utilizes physical priors to address this challenging task for the first time.
>
> **2) Correctness of equation 4.**
> Thank you for pointing out this typo. The physically correct version of the composition equation should be:
>
> $$
>   I(u, v, j) = (1 - \boldsymbol{\phi})I_t(u,v,j) + (\boldsymbol{\phi}) I_o(u,v,j)
> $$
>
> We will correct this typo in the revised version of the paper.
>
> **3) Details of the datasets.**
> Thank you for bringing this up. We will include these details in the revised paper.
>
> - The curated BDD100K dataset contains 8 scenes, while the self-collected DCVR dataset includes 10 scenes. Each sequence consists of approximately 300 frames, extracted from 10-second videos at a frame rate of 30 Hz.
> - Modern dash cameras provide decent quality. The resolution of the images in BDD100K is around 1296x500. For the DCVR dataset, we used a 4K wide-angle camera (purchased on Amazon, selected based on high sales volume). The original resolution of these images is 3840x2160. We performed camera calibration and undistortion using a chessboard and the OpenCV library before running the SFM algorithm. To facilitate the training process, we downsampled the images 3 times to a resolution similar to that in BDD100K, aligning with popular datasets like KITTI-360 and BDD100K.
> - 4 out of 8 scenes in BDD100K contain moderate dynamic objects. No scenes in DCVR contain dynamic objects.
> - As shown in Fig. 7 in the paper, our dataset includes opaque occlusions. Our dataset does not contain car hoods, which we will discuss later.
> - 3 out of 8 scenes in BDD100K contain turning. 4 out of 10 scenes in DCVR contain turning. We visually depict the turning trajectories in Fig. 2 of the attached PDF.
>
>
> **4) How well the proposed method works for the reflective car hoods?**
> Thank you for bringing up the topic of car hoods. While car hoods are indeed prone to strong reflections, they typically occupy only around 10\% of the lower parts of the images, which do not contain much information about the street scenes. Additionally, car hoods are consistently located in the same position in each video. We addressed this issue by manually cropping out the areas containing the car hoods during the dataset curation process. We believe that this can still be done on a large scale, as segmenting car hoods is not a particularly difficult task.
>
> **5) Limitations.** Limitations are discussed in the global response.

---

> ### Comment · Reviewer_zoS8 · 2024-08-11
>
> Thanks for your answers to my questions. I am glad to know more details about the evaluated data. These evaluated data are sufficiently diverse to demonstrate the improvements brought by the proposed method.
>
> I am happy with the authors' response and will remain or possibly raise my score later.
>
> One additional concern I had is similar to the Reviewer si9q's W1, the proposed method is practically limited to the car-mounted dash cam, but it doesn't address the obstruction challenge from the general scenes. This is still a key limiting point.

---

> > ### Author Response · Authors · 2024-08-12
> > **Thanks for the Response**
> >
> > We are very happy to hear that you are satisfied with our responses.
> > We would like to kindly clarify that our method is specifically designed for autonomous driving scenarios not for general scenes. Dash cameras are widely used in vehicles and  dash cam footage offers unique value for autonomous driving.
> > Compared to general scenes, dash camera in vehicles contain some important properties that can be leveraged to tackle reflections and obstructions. As shown in main paper, the obstruction removal works designed for general scenes show quite worse results, since this task is quite ill-posed and achieving good results in specific scenes is also not that easy.  Thanks for your suggestions, we would further study this task for the goal of achieving good results in broader scenes.

---

### Official Review · Reviewer_N7Lh · 2024-07-10

**Soundness:** 2
**Presentation:** 2
**Contribution:** 2
**Rating:** 4
**Confidence:** 4

**Summary:**

This paper deals with the problem of novel view synthesis in outdoor scenes captured with a dashcam. The authors develop a 3D Gaussian Splatting based method that is robust to common obstructions observed in dashcam videos, mainly due to the way these videos are captured: Mobile-phone holders, reflections and windshield stains create artifacts in the reconstruction as they are not natively modeled by 3DGS and cannot be reconstructed as part of the static 3D scene as they move with the ego-vehicle. The authors propose a learnable opacity map that decomposes the images into obstructions and real scene and a lighting modulation module that adapts obstructions over for each view based on varying illumination across views. These are implemented with a 2D hashgrid and MLP heads. They can then use the opacity map to blend the obstruction and real scene for rendering each view. The authors also use an MVS method with geometric consistency filtering to initialize the 3D Gaussian positions.

**Strengths:**

- The proposed decomposition scheme is effective in separating obstruction from real scene as illustrated in Fig. 4/5/6.
- The accompanying video results are impressive
- The different modules are clearly ablated in Tab. 2

**Weaknesses:**

- There is no description of $\mathcal{L}_{opacity}$, but without this it is not possible to judge whether the learned opacity map is learned in a self supervised manner or if additional labels are needed.
- The task setup is similar in spirit to NeRF-in-the-wild. Therefore, to judge the importance of the proposed design opposed to generic in-the-wild methods (NeRF-W,  not tailored towards dashcam videos, it would be important to understand how such methods compare on the benchmarks the authors use. However, the only baselines the authors compare to are ZipNeRF, 3DGS and GaussianPro, all of which are not designed for this type of captures.
- As far as I understand, G3E consists of running PatchMatchNet with a standard geometry consistency filter. There seems to be no innovation in that to me. What is Fig. 8 comparing exactly? Random initialization vs MVS or SfM vs MVS initialization?

**Questions:**

It would be great if the authors can clarify my concerns w.r.t. the loss details, G3E and may provide more insights on the choice of baselines.

**Limitations:**

The limitation discussion is short and does not point to meaningful failure cases.

---

> ### Author Rebuttal · Authors · 2024-08-06
>
> **1) Details of $\boldsymbol{\mathcal{L}_{opacity}}$.** The formulation of opacity loss is shown in the global response. The opacity map is learned in a self-supervised way without relying on additional labels.
>
> **2) Why only compare with general Novel View Synthesis methods?**
>
> As the first work addressing the novel task of NVS for dash cam videos, there are no prior works specifically designed for this task. As a result, we compare ours with several SOTA methods.
> NeRF-W is designed for NVS using images captured with different cameras and at different times but still cannot handle obstructions like reflections on the windshield. We evaluate NeRF-W on the dash cam videos, and as shown in the table below, our method significantly outperforms NeRF-W in both synthesis quality and rendering speed.
>
> Table 2. Comparison with NeRF-W. Tested on BDD100K
> | Method             | fps $\uparrow$ | PSNR $\uparrow$ | SSIM $\uparrow$ | LPIPS $\downarrow$ |
> |--------------------|:----------------:|:-----------------:|:-----------------:|:--------------------:|
> | Nerf in the Wild   |       0.18       |       22.58       |       0.708       |         0.395        |
> | DC-Gaussian (Ours) |        120       |       29.44       |       0.914       |         0.143        |
>
> Furthermore, as illustrated in Figure 4 in the attached pdf, although NeRF-W is designed to handle illumination variance and transient objects, it fails to separate obstructions from the images. None of the obstructions are accurately represented in the transient image. This suboptimal result occurs because the obstructions move with the camera rather than being transient parts of the images. The reflections and transmissions are intertwined in a complex manner that NeRF-W's design cannot effectively address, unlike our method.
>
>
> **3) Novelty of G3E.**
> We use the standard geometry consistency filter from PatchMatchNet to improve novel view synthesis.
> Some works also leverage MVS priors to help NVS task.
> However, simply applying PatchMatchNet to original input images does not yield optimal results. In our method, we first run DC-Gaussian without multi-view stereo (MVS) initialization. We then use the trained model to synthesize $\hat{I}_t$, as explained in Equation 7. This strategy effectively suppresses obstructions in the original images and enhances overall performance. In Fig 8, we compare the results using MVS initialization (a w/ G3E) and that using SfM initialization (b w/o G3E), we will make it more clear in revised paper.
>
> **4) Limitations.** The limitations are discussed in the global response.

---

> ### Author Response · Authors · 2024-08-13
> **Follow-Up on Rebuttal Discussion**
>
> We value your feedback and are eager to address any further questions or concerns you may have. If you have had a chance to review our response and have additional thoughts, we would greatly appreciate your input.

---

> ### Author Response · Authors · 2024-08-14
> **Awaiting Your Response as Deadline Nears**
>
> As the deadline approaches, I am eagerly anticipating your response.

---

### Official Review · Reviewer_si9q · 2024-07-12

**Soundness:** 3
**Presentation:** 3
**Contribution:** 3
**Rating:** 5
**Confidence:** 4

**Summary:**

This paper focuses on using dash cam videos for 3D Gaussian Splatting-based outdoor scene reconstruction. To address challenges such as reflections and occlusions on windshields, DC-Gaussian introduces an adaptive image decomposition module to model these effects in a unified manner. Additionally, an illumination-aware obstruction modeling technique is designed to manage reflections and occlusions under varying lighting conditions. Finally, the authors employ a geometry-guided Gaussian enhancement strategy to improve rendering details by incorporating additional geometry priors. Extensive experiments on self-captured and public dash cam videos verify the effectiveness of the proposed method.

**Strengths:**

(1) The proposed method exhibits promising 3DGS-based reconstruction performances on some self-captured and public dash cam videos.

(2) In general, this manuscript is well-structured and the writing is clear. The authors have effectively organized their ideas, making the content easy to follow and understand.

**Weaknesses:**

(1) Limited Practicality: My main concern is the practical value of this work. Dash cam videos generally provide single-view sequences, and these sparse views would result in the learned 3DGS suffering from limited novel-view synthesis quality. Additionally, the inherent limitations of dash cam videos, such as reflections and occlusions on the windshields, would further degrade rendering quality despite the authors’ efforts to mitigate these issues with developed modules. Therefore, I find it hard to believe that people would choose dash cam videos for high-quality 3D scene reconstruction in real-world applications.

(2) The authors introduce global-shared hash encoding and MLPs for illumination-aware obstruction modeling. The reviewer is curious about the running speed compared to the baseline method, 3DGS. Please provide a comparison of the inference times.

**Questions:**

See weaknesses.

**Limitations:**

The authors have adequately addressed the limitations and the potential negative societal impact of their work.

---

> ### Author Rebuttal · Authors · 2024-08-06
>
> **1) Concern about the practical value of using dash cam videos for Novel View Synthesis**
>
> Dash cam videos have unique values for autonomous driving. Dash cam videos deeply reflect the diversity and complexity of real-world traffic scenarios. They are used to provide large-scale, diverse driving video datasets in a crowd-sourced manner [1]. Dash cam videos also offer important data sources about multi-agent driving behaviors [2] and evaluating the robustness of algorithms under visual hazards [6]. BDD100K [1], a large scale diverse driving videos dataset built with dash cam videos, has been cited more than 2000 times. It has been widely used
> in many important vision tasks, such as Object Detection, Semantic Instance Segmentation [8], and Multiple Object Tracking and Segmentation [7], etc. This fact shows the huge interests of vision community in dash camera studies.
>
> Although most dash cam videos are monocular and prone to obstructions, we believe studying how to leverage the important data source is quite important to many vision tasks, such as novel view synthesis and 3D reconstruction.
> Moreover, our work is an attempt to tackle obstruction removal from dash cam videos, making it possible to achieve high-quality NVS from dash cam videos. It will provide huge 3D information for autonomous driving.
>
> [1] Yu F, Chen H, Wang X, et al. Bdd100k: A diverse driving dataset for heterogeneous multitask learning. CVPR, 2020.
>
> [2] Chandra R, Wang X, Mahajan M, et al. Meteor: A dense, heterogeneous, and unstructured traffic dataset with rare behaviors. ICRA, 2023.
>
> [3] Grand View Research. Dashboard camera market size, share \& trends analysis report by technology (basic, advanced, smart), by product, by video quality, by application, by distribution channel, by region, and segment forecasts, 2024 - 2030, 2023. Accessed: 2024-05-16.
>
> [4] Martin-Brualla, Ricardo, et al. "Nerf in the wild: Neural radiance fields for unconstrained photo collections." CVPR, 2021.
>
> [5] Gao, Chen, et al. "Dynamic view synthesis from dynamic monocular video." ICCV, 2021.
>
> [6] Oliver Zendel, Katrin Honauer, Markus Murschitz, Daniel Steininger, and Gustavo Fernandez Dominguez. Wilddash-creating hazard-aware benchmarks. In Proceedings of the European Conference on Computer Vision (ECCV), pages 402–416, 2018.
>
> [7] Luiten, Jonathon, et al. "Hota: A higher order metric for evaluating multi-object tracking." International journal of computer vision 129 (2021): 548-578.
>
> [8] Yan, Bin, et al. "Universal instance perception as object discovery and retrieval." Proceedings of the IEEE/CVF Conference on Computer Vision and Pattern Recognition. 2023.
>
> **2) Rendering speed.** This question is answered in the global rebuttal response.

---

> > ### Comment · Reviewer_si9q · 2024-08-12
> >
> > Thank you for your feedback. However, this rebuttal still does not alleviate my concerns about its practicality, especially for the novel-view synthesis (NVS):
> >
> > I acknowledge the potential application value of dashcams in many visual tasks (as the authors have listed), but this does not mean it remains significant in 3D autonomous driving scene reconstruction tasks.
> >
> > Dashcams capture monocular video, and in autonomous driving scenarios, the viewpoints are typically very sparse, inherently unsuitable for NVS. Although the authors have mitigated obstruction interference, synthesizing novel views is also crucial for 3D scene reconstruction. The authors show so-called NVS in Fig. 1 (c); however, this is somewhat misleading. From my observation, the viewpoints in Fig. 1 (b) and (c) are the same, with only obstruction removal being performed, which does not qualify as NVS. Based on my experiments, novel view synthesis from monocular autonomous driving video is very challenging. Could you present your actual NVS results? I suspect they might not be very strong.
> >
> > Finally, since autonomous driving scene reconstruction is inherently a highly challenging task, I still find it hard to believe that anyone would choose dash cam videos for high-quality 3D scene reconstruction in real-world applications.

---

> > > ### Author Response · Authors · 2024-08-12
> > > **Thanks for the Response**
> > >
> > > Thanks very much for your further response.
> > >
> > > **“The viewpoints of Fig.1 (b) and (c) are the same, which are the not the results of novel view synthesis (NVS)”**
> > >
> > > We respectfully disagree with this statement. The viewpoints of (b) and (c) must be identical because (b) represents the real-captured image used as a reference, while (c) is the synthesized image produced by our method after obstruction removal. In the context of novel view synthesis, it is standard practice to present synthesized images from a testing viewpoint alongside the corresponding real-captured reference images. Consistent with prior works in this field, we divided the captured images into two sets: the 'train split,' which is used to optimize Gaussian splatting, and the 'test split,' which is excluded from the optimization process but utilized for evaluating the synthesized images. For additional novel view synthesis results, please refer to our supplementary materials, where we also provide videos
> > >
> > > **“monocular video is inherently unsuitable for NVS”**
> > >
> > > We acknowledge that novel view synthesis (NVS) from monocular video is a challenging task, but we believe this challenge should not deter our community from making efforts to address it. In fact, NVS and 3D reconstruction from monocular cameras are long-standing problems in computer vision and computer graphics. Even in recent works related to NeRF and 3DGS, numerous studies have focused on using monocular cameras as input, such as [1,2,3,4,5,6,7,8,9,10,11]. Among these, [1, 2] are specifically designed for autonomous driving (AD) scenarios.
> > >
> > > In addition, with the rapid advancements in the computer vision community, more challenging tasks, such as generating an entire driving sequence from a single image, have become possible through recent works [12, 13]. Notably, GenAD [13], trained on monocular videos from YouTube, leverages a substantial amount of dash cam footage. This not only underscores the significant value of dash cam videos but also motivates us to explore novel view synthesis (NVS) using less constrained data sources, such as monocular dash cam videos.
> > >
> > >
> > > [1] Zhou, Hongyu, et al. "Hugs: Holistic urban 3d scene understanding via gaussian splatting." Proceedings of the IEEE/CVF Conference on Computer Vision and Pattern Recognition. 2024.
> > >
> > > [2] Lu, Fan, et al. "Urban radiance field representation with deformable neural mesh primitives." Proceedings of the IEEE/CVF International Conference on Computer Vision. 2023.
> > >
> > > [3] Gao, Chen, et al. "Dynamic view synthesis from dynamic monocular video." ICCV, 2021.
> > >
> > > [4] Wu, Tianhao, et al. "D^ 2NeRF: Self-Supervised Decoupling of Dynamic and Static Objects from a Monocular Video." Advances in neural information processing systems 35 (2022): 32653-32666.
> > >
> > > [5] Tian, Fengrui, Shaoyi Du, and Yueqi Duan. "Mononerf: Learning a generalizable dynamic radiance field from monocular videos." Proceedings of the IEEE/CVF International Conference on Computer Vision. 2023.
> > >
> > > [6] Das, Devikalyan, et al. "Neural parametric gaussians for monocular non-rigid object reconstruction." Proceedings of the IEEE/CVF Conference on Computer Vision and Pattern Recognition. 2024.
> > >
> > > [7] You, Meng, and Junhui Hou. "Decoupling dynamic monocular videos for dynamic view synthesis." IEEE Transactions on Visualization and Computer Graphics (2024).
> > >
> > > [8] Tretschk, Edgar, et al. "Non-rigid neural radiance fields: Reconstruction and novel view synthesis of a dynamic scene from monocular video." Proceedings of the IEEE/CVF International Conference on Computer Vision. 2021.
> > >
> > > [9] Fu, Yang, Ishan Misra, and Xiaolong Wang. "MonoNeRF: learning generalizable NeRFs from monocular videos without camera poses." International Conference on Machine Learning. PMLR, 2023.
> > >
> > > [10] Park, Byeongjun, and Changick Kim. "Point-dynrf: Point-based dynamic radiance fields from a monocular video." Proceedings of the IEEE/CVF Winter Conference on Applications of Computer Vision. 2024.
> > >
> > > [11] Wang, Fusang, et al. "Planerf: Svd unsupervised 3d plane regularization for nerf large-scale scene reconstruction." 3DV (2023).
> > >
> > > [12] Blattmann, Andreas, et al. "Align your latents: High-resolution video synthesis with latent diffusion models." Proceedings of the IEEE/CVF Conference on Computer Vision and Pattern Recognition. 2023.
> > >
> > > [13] Yang, Jiazhi, et al. "Generalized predictive model for autonomous driving." Proceedings of the IEEE/CVF Conference on Computer Vision and Pattern Recognition. 2024.

---

> > > > ### Comment · Reviewer_si9q · 2024-08-12
> > > >
> > > > Thank you to the authors for the response and clarification regarding Fig. 1. I realize I had misunderstood Fig. 1. After reading the authors' response and other reviewers' comments, I have started to appreciate the potential of dash cam videos in autonomous driving scenarios, although I think there are still many practical challenges, such as the limited NVS capabilities in sparse viewpoints (e.g., large rotational angle changes) and the issue of mismatched image properties (e.g., resolution) among different models of dashcams. Furthermore, as a new task, the original manuscript lacks a detailed introduction to the benchmark datasets (e.g., which sequences were selected an and the criteria for selection), which is crucial for performance evaluation. Although there have been some additions in the rebuttal, it still detracts from the paper's credibility. I suggest that the authors address this in the final draft. Overall, considering the feedback from other reviewers and the authors' responses, I am willing to raise my score in my final decision.

---

> > > > > ### Author Response · Authors · 2024-08-13
> > > > > **Thanks for the Response**
> > > > >
> > > > > Thank you for your response. I'm glad that our explanation helped clarify your confusion. I also greatly appreciate your suggestion regarding the dataset introduction. We will certainly incorporate this feedback in the final draft.

---

### Author Rebuttal · Authors · 2024-08-06

We are grateful to all reviewers for their insightful and constructive suggestions. We are glad that reviewers found: (1) The problem setting is novel (Reviewer QCPG) and meaningful (Reviewer zoS8); (2) The proposed method is interesting, smart (Reviewer zoS8), and effective (Reviewer zoS8, N7Lh); (3) The results look promising (Reviewer si9q), impressive (Reviewer N7Lh), good, and outstanding (Reviewer QCPG). Each comment from the reviewers has been replied to individually. Here we first respond to common questions.

**1) Rendering speed.** As shown in the table below, DC-Gaussian achieves 120 fps at a resolution of 1920x1080 on an RTX 3090 GPU. Although ours is slightly slower than 3DGS, the speed still enables real-time rendering, which is crucial for applications such as autonomous driving simulators.

Table 1. Rendering Speed Analysis. Tested on one NVIDIA RTX 3090 GPU with image resolution $1920\times1080$
| Method             | fps $\uparrow$ |
|--------------------|:----------------:|
| Nerf in the Wild   |       0.18       |
| ZipNeRF            |       0.27       |
| GaussianPro        |        210       |
| 3DGS               |   155  |
| DC-Gaussian (Ours) |        120       |

**2) Details of $\mathcal{L}_{opacity}$.**  The formulation of **$\mathcal{L}_{opacity}$**  is $\sum_{(u, v)}\Vert \phi(u, v) \Vert_{1}$, where $(u, v)$ are the image coordinates. This opacity loss encourages the opacity map to have the minimum areas that could satisfy the optimization. This design is based on the prior knowledge that opaque objects typically occupy only small portions of the windshield.

**3) Limitations.**
Our work focuses on obstruction removal during Novel View Synthesis (NVS) for dash cam videos. Our method is not specifically designed to improve performance on multiple sequences or dynamic scenes. These limitations have been discussed in the paper. Here, we provide additional experimental results in Figure 3 of the attached PDF. The results demonstrate that dynamic objects do not significantly impact the performance of obstruction removal. When dynamic objects move at a quite slow speed, our method also presents reasonable results. Recent efforts on modeling dynamic objects with high speed, such as StreetGaussian [1], have achieved impressive results. We plan to incorporate these techniques into our method in future research to enable robust dynamic modeling.

[1] Yan, Yunzhi, et al. "Street gaussians for modeling dynamic urban scenes." ECCV, 2024.

---

### Decision · Program_Chairs · 2024-09-25

**Decision:**

Accept (poster)

**Comment:**

As the Area Chair for submission 4323, I have thoroughly evaluated the reviewers' feedback and the authors' responses. The consensus leads me to recommend acceptance of this paper for NeurIPS 2024.

The paper addresses novel view synthesis with dash cam videos, a topic of growing importance but limited exploration. It introduces DC-Gaussian, an innovative method that effectively tackles unique challenges such as obstructions and variable lighting in dash cam footage. The experimental results are robust, showing clear improvements over existing methods like NeRF-W, especially in handling common dash cam obstructions.

Reviewers initially expressed concerns about the practicality and the handling of dynamic scenes. However, the authors provided comprehensive rebuttals that demonstrated the method's utility and robustness, supplemented by recent literature and additional experimental validations.

he authors are encouraged to incorporate final feedback regarding dataset details and benchmarking in their final manuscript. The acceptance of this paper will foster further research in a promising direction.